# *Exserohilum turcicum* Alters Phyllosphere Microbiome Diversity and Functions—Implications for Plant Health Management

**DOI:** 10.3390/microorganisms13030524

**Published:** 2025-02-27

**Authors:** Shengqian Chao, Yifan Chen, Jiandong Wu, Yin Zhang, Lili Song, Peng Li, Yu Sun, Yingxiong Hu, Hui Wang, Yuping Jiang, Beibei Lv

**Affiliations:** 1Key Laboratory of Agricultural Genetics and Breeding, Biotechnology Research Institute, Shanghai Academy of Agricultural Sciences, Shanghai 201106, China; chaoshengqian@saas.sh.cn (S.C.); chenyifan@saas.sh.cn (Y.C.); zhangyinsy1986@126.com (Y.Z.); songlili@saas.sh.cn (L.S.); sunsite@126.com (P.L.); yvessuen@hotmail.com (Y.S.); 2Key Laboratory for Safety Assessment (Environment) of Agricultural Genetically Modified Organisms, Ministry of Agriculture and Rural Affairs, Beijing 100125, China; 3Shanghai Professional Technology Service Platform of Agricultural Biosafety Evaluation and Testing, Shanghai 201106, China; 4National Engineering Laboratory of Crop Stress Resistance Breeding, School of Life Sciences, Anhui Agricultural University, Hefei 230036, China; wujiandong@ahau.edu.cn; 5CIMMYT-China Specialty Maize Research Center, Shanghai 201403, China; yxionghu@163.com (Y.H.); wanghui19840109@163.com (H.W.); 6College of Ecological Technology and Engineering, Shanghai Institute of Technology, Shanghai 201418, China

**Keywords:** *Exserohilum turcicum*, maize, metagenomic sequencing, microbiome, phyllosphere

## Abstract

The phyllosphere represents the largest biological surface on Earth and serves as an untapped reservoir of functional microbiota. The phyllosphere microbiome has the potential to mitigate plant diseases; however, limited information exists regarding its role in maintaining plant health. In this study, metagenomic sequencing was employed to analyze the microbiomes of the adaxial and abaxial leaf surfaces of healthy (CKWT) and diseased (EWT) maize, with the aim of dissecting the influence of *Exserohilum turcicum* on phyllosphere microbiome function. *E. turcicum* altered the alpha and beta diversity of the phyllosphere microbiome, with the Shannon diversity and Chao1 index values significantly reduced in EWT. More beneficial microbes accumulated in the CKWT phyllosphere, whereas pathogenic microbes decreased. *E. turcicum* may have altered the balance between commensal and pathogenic microorganisms. The species and abundances of microorganisms on the two sets of leaf surfaces were also altered after inoculation with *E. turcicum*. Further analysis of disease-resistance-related metabolic pathways and abundances of antibiotic-resistance genes revealed that *E. turcicum* altered the abundance of the functional microbiome and modified the microbiome differences between adaxial and abaxial leaf surfaces. In conclusion, the results reveal that microbial diversity in the maize phyllosphere can influence the microbiome and regulate microbial functions to support plant health. These findings enhance our understanding of how *E. turcicum* affects the phyllosphere microbiome and provide a theoretical basis for biological control of *E. turcicum*.

## 1. Introduction

Maize (*Zea mays* L.) is a widely cultivated and versatile cereal crop, accounting for approximately 40% of the world’s total cereal production [1]. Its productivity is frequently hampered by pathogens [2]. Foliar diseases threaten maize production on the continent; *Exserohilum turcicum* is one of the most devastating fungal diseases of maize, widely distributed around the world, and is the cause of northern corn leaf blight (NCLB) in maize [2,3,4]. *E. turcicum* is a facultative parasitic fungus that seriously threatens maize production. It primarily infects corn leaves, leaf sheaths, and bracts through the attached cells it produces [5,6]. NCLB has persisted for decades as a major foliar disease in maize-producing regions of the world [7,8]. The maize leaf spot disease caused by this fungus occurs in all maize-producing areas of China, with early lesions appearing as light-green water stains. As the disease progresses, these lesions expand and merge, forming spindle-shaped lesions with dark-brown edges and yellow-brown or gray-brown centers. This ultimately leads to premature wilting of the leaves, seriously affecting the photosynthetic efficiency of corn and significantly reducing both yield and quality [9,10]. This disease is particularly severe in humid and warm climatic conditions, posing a significant threat to maize yields [2]. Severe infection by NCLB (northern corn leaf blight) can result in yield losses of over 50% [11,12]. NCLB disease may also reduce the feeding value of maize and increase the incidence of maize stalk rot [13].

With the intensification of agriculture and the widespread cultivation of single varieties, the frequency and severity of maize leaf spot disease have been increasing annually, posing a major challenge to global maize production. The leaf pathogen population in tropical and subtropical regions exhibits extremely high genetic diversity, such that variety-based disease resistance is the most effective technique for controlling NCLB [2].

The aboveground parts of plants are collectively referred to as the phyllosphere, which includes leaves, flowers, stems, fruits, and pollen [14]. The surface area of the phyllosphere is approximately twice that of the land, providing a habitat for many microorganisms that settle on the surfaces of leaves and in the spaces inside leaves [15]. The microbial community in the phyllosphere is a complex and dynamic ecosystem that plays a crucial role in plant growth and health [16,17]. This microbiome not only forms a protective barrier on the leaf surface to prevent pathogen invasion, but also help plants resist pathogens through various mechanisms, such as competitive rejection, antibiotic secretion, and the induction of plant immunity [18]. The composition and function of the phyllosphere microbiome are influenced by various factors, including plant species, environmental conditions, and pathogen infection [15].

Pathogen infection can alter the microbial community in the phyllosphere. Foliar infection of maize plants by *S. turcica* can induce the enrichment of beneficial bacterial communities in the maize rhizosphere, enhancing the plants’ defense against pathogen infection [19]. The Shannon and Simpson indices of the phyllosphere bacterial communities in highly resistant hybrid maize are significantly higher than those in susceptible varieties [20]. A study showed that *Arabidopsis thaliana* exhibits different levels of resistance to powdery mildew on the adaxial and abaxial surfaces of its leaves [21]. Examination of wild barley (*Hordeum chilense*) from Chile and Argentina showed that the abaxial surfaces exhibited higher levels of resistance to invasion by three formae speciales of cereal powdery mildew fungi compared to the adaxial surfaces [22].

However, there is currently limited research on the impact of maize leaf spot disease on microbial communities, particularly regarding the differences between microbial communities on adaxial and abaxial leaf surfaces. Understanding these changes is crucial for elucidating the mechanisms of disease occurrence, exploring new biological control strategies, and enhancing crop disease resistance.

In this study, microbiome samples of the adaxial and abaxial leaf surfaces of healthy (CKWT) and diseased (EWT) maize were collected. Metagenomic sequencing was employed to comprehensively analyze the influence of *E. turcicum* on phyllosphere microbiome function. The method not only provided information on microbial community diversity and abundance, but it also revealed the distribution of functional genes in the microbial community. Differences in microbial communities between the adaxial and abaxial surfaces of CKWT and EWT maize leaves were analyzed to explore the effects of *E. turcicum* on plant–microbial interactions to identify potentially beneficial microbes in the phyllosphere and to reveal potential ecological impacts. This study can help us understand how *E. turcicum* affects the phyllosphere microbiome, identify microbial communities that favor biological resistance to the disease, and provide a theoretical basis for biological control of *E. turcicum*.

## 2. Materials and Methods

### 2.1. Experimental Design and Sampling

A pot experiment was conducted in a greenhouse. Maize seeds (KN5585) provided by the School of Life Sciences, Anhui Agricultural University, were sown in soil collected from the Baihe base (latitude: 31°24′ N, longitude: 121°11′ E). The soil was directly excavated from the field, mixed, and placed in each pot. Lab-preserved *E. turcicum* strains were used, which were transferred from glycerol stored at −80 °C to PDA medium and cultured at room temperature for 2–3 weeks. The spore suspension of *E. turcicum* was prepared by flooding each PDA plate with approximately 8 mL of ddH_2_O, which contained 0.1% Tween 20. The spores were then gently removed using a glass rod. Subsequently, approximately 5 mL of the undiluted spore suspension was transferred into each centrifuge tube. To adjust the concentration of the suspension, additional ddH_2_O was added until it reached a final concentration of 4 × 10^3^ conidia per mL [23]. An artificial inoculation procedure was carried out as described by Wu et al. [24]. A quantity of 1 mL of spore suspension of *E. turcicum* was sprayed on both the adaxial and abaxial surfaces of the leaves at the seedling stage (V4). After two weeks, the adaxial and abaxial leaf surfaces of plants at the seedling stage (V6) were sampled for the metagenomic sequencing experiment. The sampling times and positions of the several samples were the same. The leaves were the fifth and sixth leaves counted from bottom to top. The two surfaces of the leaves were from the same position on the same leaf of the same seedling. The samples were labeled as CKWT-adaxial, EWT-adaxial, CKWT-abaxial, and EWT-abaxial. CKWT represented wild maize without inoculation with *E. turcicum*, and EWT represented wild maize inoculated with *E. turcicum*. Three replicates were performed for each group, with three seedlings in each replicate. Seedlings were watered once or twice per week, depending on soil humidity, 500 mL each time.

### 2.2. Sample Collection

The sampling position on the adaxial surface of the leaves corresponded to the sampling position on the abaxial surface. A sterile cotton swab was used to randomly scrape microorganisms from the surfaces of 10 g fresh leaves into a centrifuge tube, to which 90 mL of sterile 0.1 mmol L^−1^ potassium phosphate-buffered solution (pH = 7.4) was added. The centrifuge tube was shaken in a shaker for 5 min, sonicated for 1 min, and vortexed for 10 s. The above steps were repeated twice, and then the washing solution was passed through a 0.22 μm filter membrane [25,26]. The membrane was frozen in liquid nitrogen and stored at −80 °C.

### 2.3. DNA Extraction and Sequencing

DNA extraction was performed using OMEGA’s Mag Bind Soil DNA Kit (catalog number M5635-02) (Omega Bio-Tek, Norcross, GA, USA). The concentration of extracted DNA was measured using the Qubit™ 4 Fluorometer (Invitrogen, Waltham, MA, USA), and DNA quality was assessed by 1% agarose gel electrophoresis. The standard Illumina TruSeq DNA Sample Preparation Guide was used to construct the required genomic computer library with an insertion size of ~400 bp. Each library was sequenced using the Illumina NovaSeq platform (Illumina, San Diego, CA, USA) and the PE150 strategy provided by Personal Biotechnology Co., Ltd. (Shanghai, China). All raw sequences were deposited in the National Center for Biotechnology Information (NCBI) Sequence Read Archive under accession number PRJNA1138799.

### 2.4. Bioinformatic Analysis of the Metagenomes

Raw sequencing reads were processed to obtain quality-filtered reads for further analysis. CDS sequences of all samples were clustered by mmseqs2, with a protein sequence identity threshold of 0.95 and a minimum coverage of 90% for the shorter contigs. To assess the abundances of these genes, the high-quality reads from each sample were mapped onto the predicted gene sequences using Minimap2, and the number of reads aligned to gene sequences was counted using feature counts [27]. Community diversity analysis was performed using the QIIME 2 platform, including the calculation of alpha diversity (Shannon index and Chao1 index) and beta diversity (Bray–Curtis distance). Spearman rank correlation coefficients between bacterial and fungal metagenomic data and microbial abundance were calculated using Mothur software (v.1.48.2). Relevant information was filtered with |rho| > 0.8, and a correlation network was constructed with a *p* value < 0.01 and imported into Cytoscape software (v.3.10.3) for visualization. The functionality of the non-redundant genes was obtained by annotation with mmseqs2 in “search” mode against KEGG protein databases and with Diamond (v2.0.15) against the CARD database.

## 3. Results

### 3.1. E. turcicum Influenced the Assembly of the Phyllosphere Microbiome

Sequence analysis showed that the average numbers of sequences for CKWT-adaxial, EWT-adaxial, CKWT-abaxial, and EWT-abaxial were 10,824,340,373, 10,495,960,371, 9,887,670,864, and 11,345,995,241, respectively. After quality control (QC), the average numbers of retained sequences were as follows: CKWT-adaxial = 10,443,297,718 sequences, EWT-adaxial = 66,951,601 sequences, CKWT-abaxial = 9,519,769,319 sequences, and EWT-abaxial = 72,362,067 sequences (Appendix A).

The Shannon diversities of bacteria (*p* = 0.001) and fungi (*p* = 0.001) in CKWT maize were significantly higher than those in EWT (Figure 1a,b). Principal coordinate analysis (PcoA) showed that *E. turcicum* (*p* = 0.001 for both bacteria and fungi) significantly altered the structure of bacterial and fungal communities (Figure 1c,d). Specifically, *E. turcicum* significantly reduced the Shannon diversity of both bacterial and fungal phyllosphere communities (Figure 1).

The effects of *E. turcicum* on the microbiomes of the adaxial and abaxial surfaces of the maize leaves were further analyzed. LMMs were used to identify the most important driver of microbial alpha diversity, considering both *E. turcicum* and niche effects. *E. turcicum* significantly affected the fungal Shannon diversity (*p* = 0.043) and Chao1 index values (*p* = 0.019), while niche only significantly affected the fungal Chao1 index (*p* = 0.014) (Table 1). Obviously, *E. turcicum* significantly changed the bacterial and fungal Shannon diversities for both sets of leaf surfaces (Appendix A). Interestingly, before maize inoculation with *E. turcicum*, the bacterial and fungal Shannon diversities of the adaxial surfaces of the leaves were slightly lower than those of the abaxial surfaces. However, after inoculation, the bacterial and fungal Shannon diversities of the adaxial surfaces of the leaves were significantly higher than those of the abaxial surfaces. These results suggest that *E. turcicum* changed the difference in Shannon diversity between the adaxial and abaxial surfaces of the leaves. Regarding beta diversity, the composition of bacterial and fungal communities in the maize phyllosphere was not significantly altered by niche but was significantly affected by *E. turcicum* (Appendix A; Table 2). Principal component analysis (PCA) of the identified microbial phyla revealed that *E. turcicum* changed the dominant microbiomes of both leaf surfaces. Actinobacteria, Bacteroidota, and Proteobacteria dominated the CKWT-adaxial surfaces, Basidiomycota dominated the CKWT-abaxial surfaces, *Proteobacteria* dominated the EWT-adaxial surfaces, and Ascomycota dominated the EWT-abaxial surfaces (Appendix A).

### 3.2. E. turcicum Altered the Stability of the Phyllosphere Microbiome Co-Occurrence Network

To investigate how *E. turcicum* affected co-occurrence patterns of the maize microbiomes on the adaxial and abaxial surfaces of CKWT and EWT, the bacterial–fungal interkingdom networks were analyzed. The interkingdom co-occurrence networks indicated that the network of the adaxial surfaces of leaves was more stable than that of the abaxial surfaces of leaves, and *E. turcicum* destabilized the adaxial leaf surface microbiome network (Figure 2). The modularity was higher in the CKWT networks (including the adaxial and abaxial surfaces of leaves) than in the EWT networks, while the opposite pattern was found for the average degree (Table 3). Interestingly, the proportion of negative edges was higher in CKWT-adaxial networks (the proportion of negative edges: 52.2%) compared to EWT-adaxial networks (the proportion of negative edges: 50.2%). However, in the CKWT-abaxial networks (the proportion of negative edges: 48.75%), the proportion was lower than in the EWT-abaxial networks (the proportion of negative edges: 48.81%). The results showed that *E. turcicum* altered the microbial community interactions between the two surfaces of the leaves. The ratio of fungal/bacterial nodes on the adaxial surfaces of leaves was higher than that on the abaxial surfaces of leaves in both CKWT and EWT maize (Table 3), indicating that bacterial–fungal interactions differed between the two leaf surfaces. The top 10 bacterial and fungal genera were similar across CKWT-adaxial, CKWT-abaxial, EWT-adaxial, and EWT-abaxial (Figure 2). Tilletiaria was dominant in CKWT (Figure 2).

Moreover, all the key microbiomes identified in the CKWT-adaxial co-occurrence network were bacterial, primarily belonging to *Proteobacteria* and *Actinobacteria* (Appendix A). In contrast, the key microbiomes identified in the CKWT-abaxial, EWT-adaxial, and EWT-abaxial co-occurrence networks were fungal, mainly from *Ascomycota* and *Basidiomycota* (Appendix A). Both CKWT-abaxial and EWT-abaxial shared key species, including *Aspergillus fumigatus*, *Metarhizium album*, and *Salinomyces thailandica*. These results showed that *E. turcicum* changed the key microbiomes on the adaxial surfaces of leaves, but its effect on the abaxial surfaces of leaves was less pronounced than on the adaxial surfaces of leaves (Appendix A).

### 3.3. E. turcicum Changed the Bacterial and Fungal Genera on the Adaxial and Abaxial Surfaces of Both CKWT and EWT Maize Leaves

To identify the differences in bacterial and fungal communities between the two leaf surfaces of CKWT and EWT maize, 12 dominant bacterial and fungal groups (with relative abundances ≥ 1.0%) at the species level were detected (Figure 3), and analysis of variance (ANOVA) was performed. The results showed that the relative abundances of the dominant genera varied across the different samples (Figure 3a,b). *Novosphingobium* was more abundant in EWT-adaxial and EWT-abaxial than in CKWT-adaxial and CKWT-abaxial (*p* < 0.001, ANOVA), while *Massilia* and *Hyphomicrobiales* showed the opposite trend (Figure 3c). The relative abundance of *Bradyrhizobium* was lowest in EWT-adaxial. *Pseudomonas* and *Methylobacterium* were slightly abundant in EWT-adaxial and EWT-abaxial. For fungi, *Tilletiaria* and *Basidiomycota* were more abundant in CKWT-adaxial and CKWT-abaxial than in EWT-adaxial and EWT-abaxial, whereas the opposite was true for *Ustilaginaceae* and *Ustilago* (*p* < 0.001, ANOVA) (Figure 3d). Interestingly, *Tilletiaria* and *Basidiomycota* were more abundant on the abaxial surfaces than on the adaxial surfaces of leaves, but *E. turcicum* eliminated this difference. *Moesziomyces*, *Bipolaris*, and *Penicillium* were slightly more abundant in EWT-adaxial and EWT-abaxial, whereas the opposite was observed for *Rhizopus* (Figure 3d). Taken together, these results indicate that *E. turcicum* increased the abundance of pathogens and decreased beneficial microbes in the maize phyllosphere.

### 3.4. E. turcicum Showed No Difference in the Dominant Resistance Gene Categories of the Maize Phyllosphere Microbiome

To investigate whether *E. turcicum* caused changes in the microbial-resistance genes and pathways of the maize phyllosphere (CKWT-adaxial, CKWT-abaxial, EWT-adaxial, and EWT-abaxial), the CARD and KEGG databases were screened. A total of 11 resistance genes and 8 pathways were identified (Figure 4). However, there were no differences in these genes or pathways between the four samples. *Paer_PhoQ_CST*, *Kleb_PhoP_CST*, and *Paer_PhoP_CST* were all involved in the resistance mechanism of antibiotic target alteration. Their efflux regulators were proteins and two-component regulatory systems modulating antibiotic efflux. Additionally, the pathway type of Ko02020 was a two-component system. *Mtub_katG_INH* was a catalase–peroxidase responsible for catalyzing the activation of isoniazid. *TetA(58)* is a tetracycline efflux pump involved in antibiotic efflux. The relative abundances of *tetA(58)* in the microbial communities of the four samples were relatively high compared to those of the other genes (Figure 4a).

A total of eight disease-resistance pathways were enriched: ko04626, ko02024, ko02020, ko01503, ko01502, ko01501, ko00940, and ko00280. These pathways are involved in plant–pathogen interactions, quorum sensing, two-component systems, cationic antimicrobial peptide (CAMP) resistance, vancomycin resistance, beta-lactam resistance, phenylpropanoid biosynthesis, and aromatic compound metabolism, respectively. More genes were enriched in ko00280 compared to the other pathways (Figure 4b). The abundances of these genes and pathways in EWT were slightly higher than those in WT (Figure 4). These results indicate that *E. turcicum* indeed induces a response in the maize phyllosphere microbiome to pathogens.

### 3.5. E. turcicum Altered the Abundance of Functional Microbiomes in the Maize Phyllosphere

To further investigate the potential role of the core microbiome in disease resistance, a species contribution analysis was conducted. The gene (*tetA(58)*) which had the highest abundance and the pathway (ko00280) with the most enriched genes were selected for further analysis. The relative abundances of *tetA(58)* and ko00280 in the microbial communities of the four samples (CKWT-adaxial, CKWT-abaxial, EWT-adaxial, and EWT-abaxial) presented significant differences (Figure 5). After inoculation with *E. turcicum*, both the species and proportions of the core microbiomes were obviously changed (Figure 5).

For *tetA(58)*, the relative abundances of *Methylobacterium aquaticum* and *Microbacterium* sp. p3–SID336 were significantly higher in EWT compared to CKWT, regardless of whether they were on the adaxial or abaxial leaf surfaces (Figure 5a). The relative abundances of *Arsenicibacter rosenii*, *Ideonella sakaiensis*, and *Bradyrhizobium macuxiense* were significantly higher in CKWT than in EWT. The proportion of *M. aquaticum* was also higher on both surfaces of leaves for EWT. The proportion of *Rugosimonospora africana* was also higher in EWT-adaxial and EWT-abaxial compared to CKWT-adaxial and CKWT-abaxial. The proportion of *Microbacterium* sp. p3–SID336 was significantly higher in EWT-adaxial compared to EWT-abaxial.

For ko00280, the core microbiome in CKWT was completely different from that in EWT (Figure 5b). The relative abundances of the top 20 microbial species were different between CKWT-ab and EWT-ab and between CKWT-ad and EWT-ad (Figure 5b). The relative abundances of *Bradyrhizobium macuxiense*, *Bradyrhizobium* sp. DFCI–1, *Nevskia soli*, and *Panacagrimonas perspica* were significantly higher in CKWT compared to EWT. The relative abundances of *Stutzerimonas stutzeri*, *Microbacterium* sp. p3–SID336, *Sphingomonas jinjuensis*, *M. aquaticum*, and *Sphingomonas* sp. MA1305 were significantly higher in EWT compared to CKWT. The proportions of *Stutzerimonas stutzeri* and *Microbacterium* sp. p3–SID336 were obviously higher in EWT-adaxial compared to EWT-abaxial, and *Sphingomonas jinjuensis* was obviously higher in EWT-abaxial compared to EWT-adaxial (Figure 5b). These results showed that *E. turcicum* changed the abundance of the functional microbiome and also changed the differences between the microbiomes of the adaxial and abaxial leaf surfaces.

## 4. Discussion

Maize is an important food and industrial crop; it is grown in more than 160 countries [28]. As one of the richest habitats on Earth [29], the plant phyllosphere hosts a diverse microbiome, many members of which are beneficial for alleviating both biological and abiotic stresses, as well as affecting plant growth and adaptation. Some microorganisms develop adaptive traits and are closely associated with leaves [15,30]. NCLB is a serious leaf disease of maize, causing yield losses of up to 50% [31]. The abundance and diversity of the phyllosphere microbiome in both healthy and NCLB-infected maize remain underexplored, suggesting the potential presence of beneficial yet undiscovered organisms or microbial activities that could enhance crop improvement. This study employed metagenomic methods to investigate the composition and diversity of phyllosphere microbial communities in both healthy and NCLB-affected maize plants, aiming to identify relevant microorganisms and explore potential strategies for enhancing plant health.

Bacteria associated with leaves are typically limited to a few phyla, including *Actinobacteria*, *Bacteroidetes*, *Firmicutes*, and *Proteobacteria* and are commonly shared across different plant species [32,33]. Studies have shown that variations in rhizosphere microbial communities can influence resistance to *Fusarium oxysporum f.* sp. niveum and have also confirmed that soil microbial populations play a crucial role in maintaining soil health and inhibiting plant diseases [34]. A decrease in soil microbial diversity has been associated with soil-borne plant diseases [35]. Compared to diseased maize (EWT inoculated with *E. turcicum*), healthy maize (CKWT) exhibited a higher abundance of phyllosphere-associated microorganisms (Figure 1a,b), indicating that differences in abundance may influence plant disease resistance. Moreover, in CKWT, the relative abundance of the microbiome on the adaxial surfaces of the leaves was almost the same as that on the abaxial surfaces. However, after inoculation with *E. turcicum*, the relative abundance of the microbiome on the adaxial surfaces of the leaves was significantly higher than that on the abaxial surfaces of the leaves, further confirming that microbiome abundance affects plant disease resistance. Decreased soil microbial diversity is responsible for the development of soil-borne plant diseases [34,36]. The abundance of rhizosphere microbial diversity in maize influences the efficacy of the rhizosphere microbiome in regulating microbial functions to manage and maintain plant health [37]. Increasing phyllosphere microbial abundance by applying exogenous microorganisms may help improve plant disease resistance.

Studies have shown that the diversity and abundance of the microbiome, especially the dominance of *Proteobacteria*, *Bacteroidetes*, *Firmicutes*, and *Cyanobacteria*, may result from changes in factors that affect microbial populations and distributions [38]. Our study also supported this result. The dominant representatives in the microbiomes of the four samples (CKWT-adaxial, CKWT-abaxial, EWT-adaxial, and EWT-abaxial) were *Actinobacteria*, *Bacteroidota*, *Proteobacteria*, *Basidiomycota*, and *Proteobacteria* (Appendix A). Co-occurrence network analysis further confirmed this result (Figure 2), suggesting that these microorganisms may all influence the microbial populations. A previous study found that the fungal genera *Gibberella*, *Neosartorya*, *Penicillium*, *Phaeosphaeria*, and *Nectria* were dominant in rhizosphere samples of healthy maize, while *Aspergillus*, *Neurospora*, and *Chaetomium* were more commonly associated with diseased maize and had higher abundances [37]. This was not entirely consistent with our results, possibly due to the specificity of microorganisms interacting with different ecological niches in plants.

Compared with diseased maize, most of the genera, such as *Streptomyces*, *Candidatus Solibacter*, *Conexibacter*, and *Bradyrhizobium*, existed in healthy maize, and these species all have a positive effect on maize growth [39,40]. These beneficial microorganisms can interact directly with pathogens by secreting chemical metabolites that inhibit their growth or make them non-toxic [41], stimulate the production of bactericidal substances in plants [42], induce the expression of plant defense genes [43], and directly produce antibacterial substances [44], and *Streptomyces* can produce antibiotics [45]. In this study, compared with EWT, the phyllosphere of CKWT gathered more beneficial microorganisms (Figure 3); therefore, *E. turcicum* may have altered the balance between commensal and pathogenic microorganisms. The ecological, physiological, and molecular mechanisms by which the microbiome influences disease resistance are complex and poorly understood. In tomato, the ability of the phyllosphere microbiome to increase resistance to Pseudomonas syringae depends on the nutritional status of the plant [46,47]. When protective microbes and biocontrol genera are reduced and pathogenic bacteria are increased, maize is at risk of pathogen invasion [35]. It was also found that soil proto-rhizosphere bacteria communities significantly reduced the incidence of disease and death caused by *Fusarium* or *Alternaria* in tobacco (*Nicotiana attenuata*) compared to infected plants growing in fungus-infested agricultural soil [48]. Studies have shown that the adaxial and abaxial surfaces of plant leaves have different resistances to pathogens [21], suggesting that further exploration and utilization of beneficial microorganisms in CKWT may offer a potential way to enhance plant resistance to biological stress.

Microorganisms function within communities rather than as individual species [18], and functional genes exhibit different patterns depending on taxa, with functional genes generally serving as better predictors of ecological niches [49,50,51]. The key factors influencing the assembly and structure of bacterial communities may not be “species”, but rather their specific functional genes [50]. In addition, functional genes may be more heritable, driven more by host genetic interactions [52]. Further analysis of microbial resistance genes and pathways on the adaxial and abaxial leaves of maize showed differences in the abundances of antibiotic-resistance genes and stress-resistant pathways observed across different samples (Figure 4). This might suggest a difference in the function of the microbiome on both leaf surfaces, and *E. turcicum* also changed the functional profile of the microbiome on the leaves.

There is mounting evidence that the plant innate immune system is centrally involved in regulating microbial symbionts. Interactions between the microbiome and the immune system may play a key role in the formation of beneficial plant–microbiota combinations and maintenance of microbial homeostasis [53]. We focused on the antibiotic-resistance genes with the highest abundance and the pathways enriched with more genes related to stress resistance and further analyzed the role of the microbiome in these pathways. Compared with EWT, the relative abundance of *Bradyrhizobium* on the leaf surface of maize in CKWT was higher for both *tetA(58)* and ko00280 (Figure 5). It was suggested that *Bradyrhizobium* is indeed a dominant microorganism in the phyllosphere [40], and *E. turcicum* might have a negative effect on the growth of *Bradyrhizobium* in the phyllosphere. The relative abundances of *Arsenicibacter rosenii* and *Ideonella sakaiensis* were higher in CKWT compared to EWT, with *Ideonella sakaiensis* exhibiting a higher abundance on the adaxial surfaces than on the abaxial surfaces of leaves. All species have been reported as beneficial microorganisms [54,55]. *Stutzerimonas stutzeri* was significantly higher on adaxial surfaces of EWT and has been found to enhance colonization and promote tomato seedling growth [56]. These results indicate that *E. turcicum* altered the composition of the maize phyllosphere microbiome. The differences between CKWT (healthy) and EWT (diseased) maize phyllosphere microbiomes suggest that microorganisms may develop specific, relevant functions within plant ecosystems.

## 5. Conclusions

The phyllosphere microbiome plays an indispensable role in plant health. Metagenomic sequencing was performed on the adaxial and abaxial surfaces of leaves from healthy and *E. turcicum*-inoculated maize. The results showed that *E. turcicum* altered the α and β diversity of the phyllosphere microbiome, and after maize was infected with *E. turcicum*, the composition and structure of the phyllosphere microbiome changed, with significant reductions in Shannon diversity and Chao1 index values, as well as changes in the abundance of the microbiome on both the adaxial and abaxial surfaces of leaves. It was found that microbial abundance and diversity were more correlated with the healthy maize phyllosphere and that more beneficial microorganisms and fewer pathogens accumulated on the CKWT phyllosphere; *E. turcicum* may have altered the balance between commensal and pathogenic microorganisms, which could have had a positive impact on the microbiome’s ability to regulate microbial functioning and maintain plant health. *E. turcicum* also altered the microbial abundance of disease-resistance-related metabolic pathways and the functional microbiome and altered microbiome differences between the adaxial and abaxial surfaces of the leaves. These findings are critical for further studies of the healthy maize phyllosphere microbiome for the isolation and screening of beneficial microorganisms that can enhance plant disease resistance, as well as for understanding the mechanisms by which *E. turcicum* affects the maize phyllosphere microbiome in terms of disease resistance and providing a theoretical basis for the biological control of *E. turcicum*.

## Figures and Tables

**Figure 1 microorganisms-13-00524-f001:**
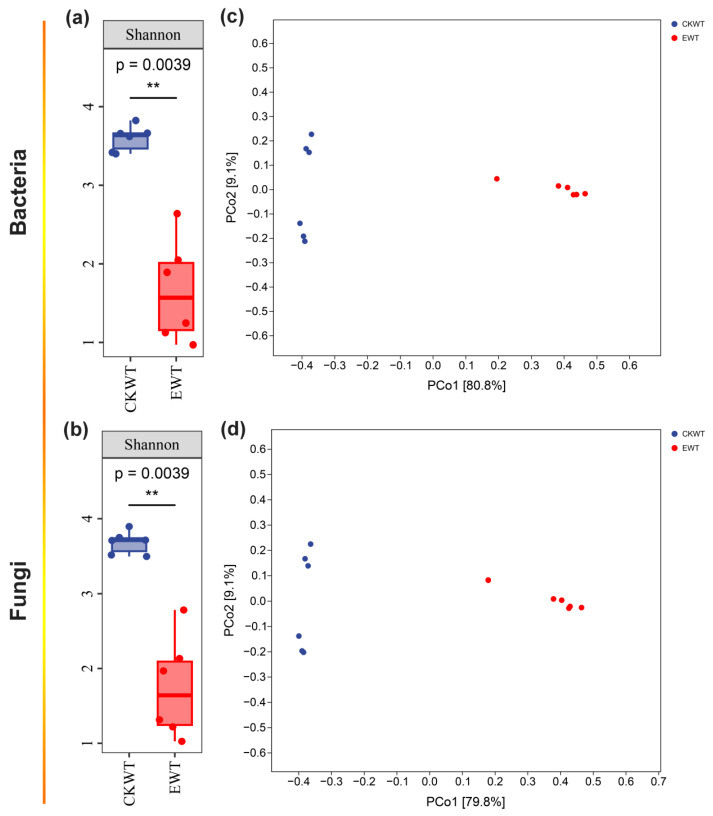
Bacterial and fungal communities in tested maize samples. Shannon index values of (**a**) bacterial and (**b**) fungal communities in the maize phyllosphere (one-way analysis of variance (ANOVA), n = 3, *p* < 0.05). Principal coordinate analysis (PcoA) ordinations based on the Bray–Curtis similarity showed differences in the structure of (**c**) bacterial and (**d**) fungal communities in the phyllospheres of wild-type (WT) and *ZmMYB3R*-overexpressing (OE) maize (permutational ANOVA (PERMANOVA), n = 3, *p* < 0.05). CKWT represents wild-type maize, and EWT represents wild-type maize inoculated with *Exserohilum turcicum*. Significant differences are marked with asterisks: **, *p* < 0.01.

**Figure 2 microorganisms-13-00524-f002:**
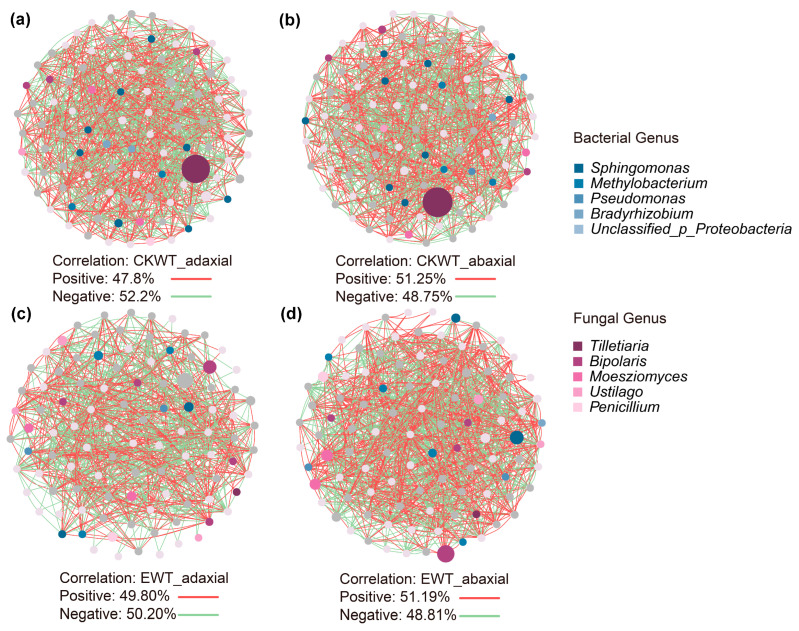
Phyllosphere microbial community co-occurrence network for (**a**) CKWT-adaxial, (**b**) CKWT-abaxial, (**c**) EWT-adaxial, and (**d**) EWT-abaxial samples. Blue nodes represent bacteria, violet nodes represent fungi, green lines represent negative correlations, and red lines represent positive correlations. The node size indicates the degree of connection. CKWT-adaxial represents adaxial surfaces of wild-type maize leaves; CKWT-abaxial represents abaxial surfaces of wild-type maize leaves; EWT-adaxial represents adaxial surfaces of wild-type maize leaves inoculated with *Exserohilum turcicum*; EWT-abaxial represents abaxial surfaces of wild-type maize leaves inoculated with *Exserohilum turcicum*.

**Figure 3 microorganisms-13-00524-f003:**
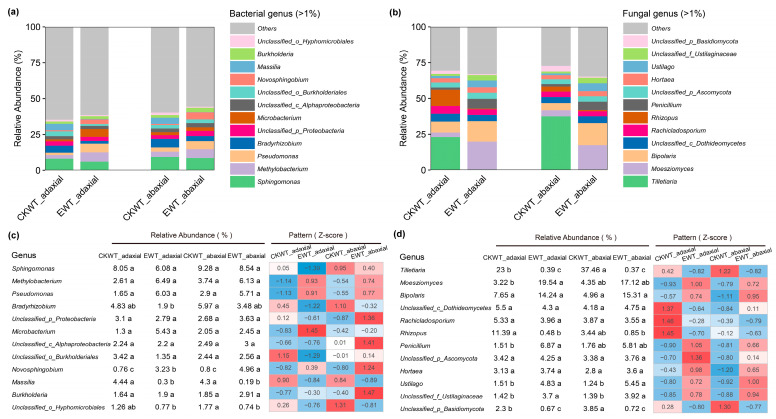
Relative abundances of dominant bacterial and fungal genera in maize phyllosphere. (**a**) Bacterial and (**b**) fungal genera with relative abundances > 0.1%. To visualize variation patterns, (**c**) bacterial and (**d**) fungal genera with relative abundances were normalized to Z-score values, where Z-score = (data point − mean)/(standard deviation). CKWT-adaxial represents adaxial surfaces of wild-type maize leaves; CKWT-abaxial represents abaxial surfaces of wild-type maize leaves; EWT-adaxial represents adaxial surfaces of wild-type maize leaves inoculated with *Exserohilum turcicum*; EWT-abaxial represents abaxial surfaces of wild-type maize leaves inoculated with *Exserohilum turcicum*. Different letters indicate significant differences between niches (n = 3, *p* < 0.001, analysis of variance (ANOVA)).

**Figure 4 microorganisms-13-00524-f004:**
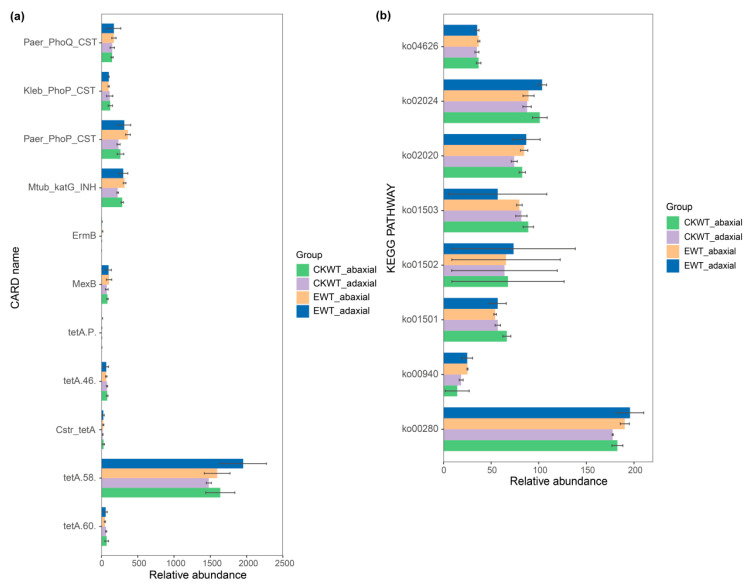
Resistance genes and pathways in the maize phyllosphere. (**a**) Relative abundances of resistance genes in the maize phyllosphere. (**b**) Disease-resistance metabolic pathways enriched by functional genes. CKWT-adaxial represents adaxial surfaces of wild-type maize leaves; CKWT-abaxial represents abaxial surfaces of wild-type maize leaves; EWT-adaxial represents adaxial surfaces of wild-type maize leaves inoculated with *Exserohilum turcicum*; EWT-abaxial represents abaxial surfaces of wild-type maize leaves inoculated with *Exserohilum turcicum*. ko04626: plant–pathogen interactions, ko02024: quorum sensing, ko02020: two-component systems, ko01503: cationic antimicrobial peptide (CAMP) resistance, ko01502: vancomycin resistance, ko01501: beta-lactam resistance, ko00940: phenylpropanoid biosynthesis, ko00280: aromatic compound metabolism.

**Figure 5 microorganisms-13-00524-f005:**
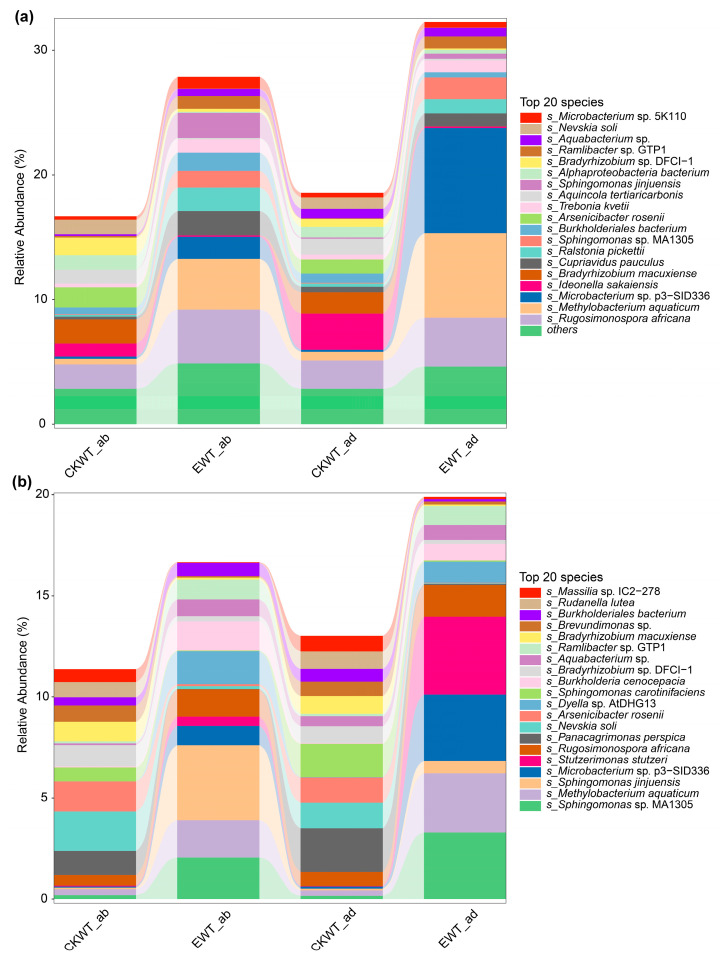
Functional genes and pathways of host microbiomes in the maize phyllosphere. (**a**) The proportion of the gene *tetA(58)* in the microbial community that influences changes in the abundance of the gene. (**b**) The proportion of pathway ko00280 in the microbial community. CKWT-adaxial represents adaxial surfaces of wild-type maize leaves; CKWT-abaxial represents abaxial surfaces of wild-type maize leaves; EWT-adaxial represents adaxial surfaces of wild-type maize leaves inoculated with *Exserohilum turcicum*; EWT-abaxial represents abaxial surfaces of wild-type maize leaves inoculated with *Exserohilum turcicum*.

**Table 1 microorganisms-13-00524-t001:** Linear mixed models (LMMs) for alpha diversity indices. The effects of genotype and niche on bacterial and fungal community alpha diversity indices were tested with LMMs. Significance was assessed using type II analysis of variance (ANOVA) with Kenward–Rodger approximation of the degrees of freedom in an LMM.

Microbial Communities	Variables	Shannon Diversity		Chao1 Index	
		*F* Value	*p* (>*F*)	*F* Value	*p* (>*F*)
Bacterial Community	*E. turcicum*	4.754	0.057	1.526	0.245
	Niche	1.997	0.191	0.433	0.527
Fungal Community	*E. turcicum*	5.574	0.043	8.144	0.019
	Niche	1.681	0.227	9.144	0.014

**Table 2 microorganisms-13-00524-t002:** Effects of maize niche and *E. turcicum* on microbial community composition assessed by permutational analysis of variance (PERMANOVA).

Microbial Community	Factor	*F* Value	R Square	*p* Value
Bacterial Community	Niche	0.3605	0.03479	0.921
	*E. turcicum*	6.876	0.40744	0.002 **
	Niche * *E. turcicum*	2.3559	0.46906	0.011 *
	Pairwise comparison			
	CKWT_adaxial vs. CKWT_abaxial	0.501442	0.111396	0.7
	CKWT_adaxial vs. EWT_adaxial	2.124	0.346832	0.1
	CKWT_abaxial vs. EWT_abaxial	5.977186	0.599085	0.1
	EWT_adaxial vs. EWT_abaxial	0.444321	0.099975	0.7
Fungal Community	Niche	0.4115	0.03952	0.767
	*E. turcicum*	15.328	0.60518	0.002 **
	Niche * *E. turcicum*	5.8321	0.68623	0.004 **
	Pairwise comparison			
	CKWT_adaxial vs. CKWT_abaxial	3.298574	0.451948	0.1
	CKWT_adaxial vs. EWT_adaxial	6.764368	0.628404	0.1
	CKWT_abaxial vs. EWT_abaxial	10.14767	0.717268	0.1
	EWT_adaxial vs. EWT_abaxial	0.290889	0.067792	0.9

CKWT-adaxial, adaxial surfaces of leaves from wild-type maize; EWT-adaxial, adaxial surfaces of leaves from wild-type maize inoculated with *Exserohilum turcicum*; CKWT-abaxial, abaxial surfaces of leaves from wild-type maize; EWT-abaxial, abaxial surfaces of leaves from wild-type maize inoculated with *Exserohilum turcicum*. Niche* *E. turcicum* represent the interactive effects of niche and *E. turcicum*. Significant differences are marked with asterisks: *, *p* < 0.05; **, *p* < 0.01.

**Table 3 microorganisms-13-00524-t003:** Topological characteristics of phyllosphere bacterial–fungal networks.

Network Indicators	CKWT-Adaxial	EWT-Adaxial	CKWT-Abaxial	EWT-Abaxial
Number of nodes	100	100	99	99
Ratio of fungal/bacterial nodes	1	1	0.98	0.98
Number of edges	862	745	843	965
Number of positive correlations	412	371	432	494
Number of negative correlations	450	374	411	471
Average length	1.493537015	1.566103669	1.491862568	1.510400812
Graph density	0.174141414	0.150505051	0.173778602	0.198928056
Network diameter	2	2	2	2
Clustering coefficient	0	0	0	0
Average degree	17.8	20.44	17.48484848	21.32323232
Modularity	0.595563654	0.516646998	0.615949646	0.412025021
Neg/pos	1.09223301	1.008086253	0.951388889	0.953441296

CKWT-adaxial, adaxial surfaces of leaves from wild-type maize; EWT-adaxial, adaxial surfaces of leaves from wild-type maize inoculated with *Exserohilum turcicum*; CKWT-abaxial, abaxial surfaces of leaves from wild-type maize; EWT-abaxial, abaxial surfaces of leaves from wild-type maize inoculated with *Exserohilum turcicum*.

## Data Availability

The raw data supporting the conclusions of this article will be made available by the authors on request.

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
