# Peer review of "Exserohilum turcicum Alters Phyllosphere Microbiome Diversity and Functions—Implications for Plant Health Management"

_microorganisms, 2025, doi:10.3390/microorganisms13030524_

Round 1

Reviewer 1 Report

Comments and Suggestions for Authors

This study was very interesting and compelling.  I do have some thoughts that I believe will help.  In the introduction in particular, I think it would be a good idea to discuss why you would want to apply a pathogen (E. turcicum)to the Maize phyllosphere.  More to the point, is there an advantage to the plant in having the pathogenic infection.  Additionally you discuss how the adaxial and abaxial surfaces of Arabidopsis have differentisal resistance to powdery mildew.  While that is true, you do not discuss why this matters towards your study.  It is hard to understand the link.  Please make this more clear in your intro.

Your materials and methods, results were well done and the figures dynamic.  Althought I owould recommend making the legend in Figure 3a larger.  And further more, in that figure you have Burkholderiales listed amongst genera.  Burkholderia is a genus.  But Burkholderiales is an order.  This is confusing.  The same for Figure 3d where Ascomycota and Basidiomycota are listed as genera.  I've had this issue with sequences as well.  You may have to go back to your raw data and sort manually to make sure that what you are stating is a genus is in fact genus.

The caption of figure five line 319-324 EWT-abxial should be changed to EWT-abaxial.  Adaxial is also misspelled in this area.  Or to be clear, say that EWT ab represents abaxial and so forth.  I really like Figure 5 but really wish your comparisons were CKWT ab vs. EWT ab, then CKWT ad vs. EWT ad because it seems your study is really trying to compare the impact of the presence of Exserophilum turcicum on the surfaces. I have the same comment throughout your results. 

In the discussion, line 327, you need to add the work Maize.  This is the first sentence of the discussion. 

Author Response

Comments 1: In the introduction in particular, I think it would be a good idea to discuss why you would want to apply a pathogen (E. turcicum) to the Maize phyllosphere.  More to the point, is there an advantage to the plant in having the pathogenic infection.

Response 1: Thanks for your careful comments.

Foliar diseases threaten maize production on the continent, E. turcicum is one of the most devastating fungal diseases of maize, widely distributed around the world, and is the cause of Northern corn leaf blight (NCLB) in maize. NCLB has persisted for decades as a major foliar disease in maize producing regions of the World (Savary et al., 2019; Nsibo et al., 2024). Studying the impact of E. turcicum on phyllosphere can help us understand how plants respond or adapt to pathogen stress through immune responses or adaptation mechanisms. Studies have shown that in some cases, the presence of pathogens may activate plant defense mechanisms, enhancing their resistance to other environmental stresses such as drought or salinity (Boller et al., 2009). The maize leaf spot disease caused by this fungus occurs in all maize-producing areas of China. This disease is particularly severe in humid and warm climate conditions, posing a significant threat to maize yield (Galiano et al., 2017). Severe infection by NCLB (Northern corn leaf blight) can result in yield loss of over 50% (Raymundo et al., 1981; Sibiya et al., 2012). Therefore, the application E. turcicum is not only used to study the interaction between pathogens and plants, but also to reveal how plants adapt to different living environments by regulating defense mechanisms under pathogen pressure.

Please see our revised manuscript for details (Page 2, and line 51-52, line 56-57, line 84-91).

References:

Savary, S., Willocquet, L., Pethybridge, S. J., Esker, P., McRoberts, N., Nelson, A. (2019). The global burden of pathogens and pests on major food crops. Nat. Ecol. Evol. 3 (3), 430–439. doi: 10.1038/s41559-018-0793-y.

Nsibo DL, Barnes I, Berger DK. Recent advances in the population biology and management of maize foliar fungal pathogens Exserohilum turcicum, Cercospora zeina and Bipolaris maydis in Africa. Front Plant Sci. 2024 Aug 1;15:1404483. doi: 10.3389/fpls.2024.1404483.

Boller T, He SY. Innate immunity in plants: an arms race between pattern recognition receptors in plants and effectors in microbial pathogens. Science. 2009 May 8;324(5928):742-4. doi: 10.1126/science.1171647.

Galiano-Carneiro AL, Miedaner T. Genetics of Resistance and Pathogenicity in the Maize/Setosphaeria turcica Pathosystem and Implications for Breeding. Front Plant Sci. 2017 Aug 29;8:1490. doi: 10.3389/fpls.2017.01490.

Raymundo A. D., Hooker A. L. Measuring the relationship between northern corn leaf blight and yield losses. Plant Dis. 1981; 65 325–327. doi: 10.1094/pd-65-325

Sibiya, J., Tongoona, P., & Derera, J. Combining ability and GGE biplot analyses for resistance to northern leaf blight in tropical and subtropical elite maize inbred lines. Euphytica, 2012; 191(2), 245-257. doi: 10.1007/s10681-012-0806-x

Comments 2: Additionally you discuss how the adaxial and abaxial surfaces of Arabidopsis have differentisal resistance to powdery mildew.  While that is true, you do not discuss why this matters towards your study.  It is hard to understand the link.  Please make this more clear in your intro.

Response 2: Thanks for your careful comments.

This paragraph mainly describes the effects of pathogens on the plant phyllosphere microbiome. We consulted relevant study and added it. Foliar infection of maize plants by S. turcica can induce the enrichment of beneficial bacterial communities in the maize rhizosphere, enhancing the plant's defense against pathogen infection (Zhu et al, 2021). The Shannon and Simpson indices of the phyllosphere bacterial communities in highly resistant hybrid maize are significantly higher than those in susceptible one (Tian et al, 2020).

Because our study also found that E. turcicum changed the community composition and structure of microbiome on the adaxial and abaxial surfaces of maize leaves. In introduction, we hope to explain why we are concerned that E. turcicum can change microorganisms on the adaxial and abaxial surfaces of maize leaves. Therefore, we have consulted many studies, but we did not find any studies on the specific analysis of the effects of pathogens on the microbiome of the adaxial and abaxial leaves of maize, and could only describe relevant studies in other species.

Please see our revised manuscript for details (Page 2, and line 84-91).

References:

Zhu L, Wang S, Duan H, Lu X. Foliar pathogen-induced assemblage of beneficial rhizosphere consortia increases plant defense against Setosphaeria turcica. Front Biosci (Landmark Ed). 2021 Sep 30;26(9):543-555. doi: 10.52586/4966.

Xueliang T, Dan X, Tingting S, Songyu Z, Ying L, Diandong W. Plant resistance and leaf chemical characteristic jointly shape phyllosphere bacterial community. World J Microbiol Biotechnol. 2020 Aug 16;36(9):139. doi: 10.1007/s11274-020-02908-0.

Comments 3:Your materials and methods, results were well done and the figures dynamic. Althought I would recommend making the legend in Figure 3a larger.  And further more, in that figure you have Burkholderiales listed amongst genera.  Burkholderia is a genus.  But Burkholderiales is an order.  This is confusing.  The same for Figure 3d where Ascomycota and Basidiomycota are listed as genera.  I've had this issue with sequences as well.  You may have to go back to your raw data and sort manually to make sure that what you are stating is a genus is in fact genus.

Response 3: Thank you for pointing this out. We have enlarged the legend of Figure 3a and 3b. And there was an error in our labeling, as it does not have a specific classification at the genus level, but belongs to the Burkholderiales at the order level or belongs to Ascomycota and Basidiomycota at the phylum level. We have revised the Figure 3.

Please see our revised manuscript for details (Page 12, line 290)

Comments 4: The caption of figure five line 319-324 EWT-abxial should be changed to EWT-abaxial. Adaxial is also misspelled in this area.  Or to be clear, say that EWT ab represents abaxial and so forth.

Response 4: Thank you for pointing this out. We have revised.

Please see our revised manuscript for details (Page 16, line 371-374)

Comments 5: I really like Figure 5 but really wish your comparisons were CKWT ab vs. EWT ab, then CKWT ad vs. EWT ad because it seems your study is really trying to compare the impact of the presence of Exserophilum turcicum on the surfaces. I have the same comment throughout your results.

Response 5: Thank you for pointing this out. We have revised the Figure 5. We also have revised the descriptions of the results regarding the comparison between CKWT-ab and EWT-ab, and the comparison between CKWT-ad and EWT-ad.

Please see our revised manuscript for details (Page 14, line 344-355; Page 16, line 367)

Comments 6: In the discussion, line 327, you need to add the work Maize.  This is the first sentence of the discussion.

Response 6: Thank you for your kind suggestions. We have added.

Please see our revised manuscript for details (Page 17, line 377)

Reviewer 2 Report

Comments and Suggestions for Authors

The manuscript, titled "Exserohilum turcicum alters phyllosphere microbiome diversity and functions, implications for plant health management," examines the interaction of the pathogen of Northern Corn Leaf Blight (NCLB- a disease of great global importance) with the foliar microbiome. The microbial communities found in the phyllosphere may play a very important role in the occurrence of infections and in the possible development of biological control technologies. These microbiomes not only form a protective barrier on the leaf surface to prevent pathogen invasion, but also help plants resist pathogens through various mechanisms, such as competitive rejection, antibiotic secretion, and induction of plant immunity. However, the composition and function of the phyllosphere microbiome are influenced by many factors, including plant species, environmental conditions, and pathogen infection.

The introduction is sufficiently detailed, well presenting the international significance and occurrence of the pathogen. The material and methods chapter is well developed. Based on the descriptions, the tests and analyses can be reproduced with sufficient accuracy.

The presentation of the results is spectacular. Illustrated with clear figures and tables.
They refer to a sufficient number of international sources (49) for comparison with their own results.

In summary, they concluded that the phyllosphere microbiome plays an essential role in effective plant protection. They performed metagenomic sequencing on the adaxial and abaxial surfaces of healthy (CKWT) and E. turcicum-inoculated (EWT) maize leaves. Their results showed that after maize was infected with the NCLB pathogen, the composition of the phyllosphere microbiome changed, significantly reducing Shannon diversity and Chao1 indices, and changing the abundance of the microbiome on the adaxial and abaxial surfaces of the leaves. Further investigation of the healthy maize phyllosphere microbiome is essential to isolate beneficial microorganisms that can enhance the plant's resistance to diseases. This research may also provide a theoretical basis for biological control against E. turcicum.

Based on the above, I recommend publishing the manuscript as a scientific article.

Author Response

Comments: The manuscript, titled "Exserohilum turcicum alters phyllosphere microbiome diversity and functions, implications for plant health management," examines the interaction of the pathogen of Northern Corn Leaf Blight (NCLB- a disease of great global importance) with the foliar microbiome. The microbial communities found in the phyllosphere may play a very important role in the occurrence of infections and in the possible development of biological control technologies. These microbiomes not only form a protective barrier on the leaf surface to prevent pathogen invasion, but also help plants resist pathogens through various mechanisms, such as competitive rejection, antibiotic secretion, and induction of plant immunity. However, the composition and function of the phyllosphere microbiome are influenced by many factors, including plant species, environmental conditions, and pathogen infection.

The introduction is sufficiently detailed, well presenting the international significance and occurrence of the pathogen. The material and methods chapter is well developed. Based on the descriptions, the tests and analyses can be reproduced with sufficient accuracy.

The presentation of the results is spectacular. Illustrated with clear figures and tables.

They refer to a sufficient number of international sources (49) for comparison with their own results.

In summary, they concluded that the phyllosphere microbiome plays an essential role in effective plant protection. They performed metagenomic sequencing on the adaxial and abaxial surfaces of healthy (CKWT) and E. turcicum-inoculated (EWT) maize leaves. Their results showed that after maize was infected with the NCLB pathogen, the composition of the phyllosphere microbiome changed, significantly reducing Shannon diversity and Chao1 indices, and changing the abundance of the microbiome on the adaxial and abaxial surfaces of the leaves. Further investigation of the healthy maize phyllosphere microbiome is essential to isolate beneficial microorganisms that can enhance the plant's resistance to diseases. This research may also provide a theoretical basis for biological control against E. turcicum.

Based on the above, I recommend publishing the manuscript as a scientific article.

Response:

Thanks for the kind and insightful comments. The phyllosphere microbiome plays an indispensable role in plant health. But there is currently limited research on the impact of maize leaf spot disease on the microbial community, particularly regarding the differences between microbial communities on the adaxial and abaxial leaf surfaces. Microbial abundance and diversity were more correlated with the healthy maize phyllosphere, which positively influenced the microbiome’s ability to regulate microbial function and maintain plant health. These results are also of great significance to elucidate the mechanism of disease occurrence, explore new biological control strategies, and improve crop disease resistance.

Reviewer 3 Report

Comments and Suggestions for Authors

The study of the phyllosphere represent a hot topic for the better understanding of plant specific microbiome. The analysis of differences between the microbiomes of healthy and diseased leaf is important for multiple research and applications purposes. Inoculation of plants with beneficial microorganism can be designed to improve their leafs and thus the survival in different pathogen attack degrees.

Some changes can be done to the current form of the manuscript to improve the presentation of the findings.

Introduction section - the background and necessity of the study is well presented. The authors should write at the end of this section a separate paragraph where to present a clear aim of this study and the hypotheses proposed for the research.

Materials and methods section is clear, presenting all the information necessary to understand the study, the experimental design and the type of analysis performed. The authors used multiple data analysis to explore their findings.

The entire Results section is well organized in small and concise sub-sections. All the results are explored, and the text associated with tables and figures is of an appropriate length. 

The Discussion section explore the main trends observed by the authors and make connections with multiple international references in the field. This section complete well the manuscript and bring multiple perspectives for future researches.

The Conclusion section needs to be modified to point more specific to the most important findings of this study. In the current form is too general.

Author Response

Comments 1: Introduction section - the background and necessity of the study is well presented. The authors should write at the end of this section a separate paragraph where to present a clear aim of this study and the hypotheses proposed for the research.

Response 1: Thanks for your kind comments. We have added a separate paragraph at the end of introduction.

In this study, microbiome samples of the adaxial and abaxial leaf surfaces of of healthy (CKWT) and diseased (EWT)·maize were collected. Metagenomic sequencing was employed to comprehensively analyze the influence of E. turcicum on phyllosphere microbiome function. The method not only provided information on microbial com-munity diversity and abundance, but also revealed the distribution of functional genes in the microbial community. Differences in microbial communities between the adaxial and abaxial surfaces of CKWT and EWT maize leaves were analyzed to explore the effects of E. turcicum on plant-microbial interactions, to identify potentially beneficial microbes in the phyllosphere, and to reveal potential ecological impacts. This study can help us understand how E. turcicum affectd phyllosphere microbiome, identify microbial communities that favor biological resistance to the disease, and provide a theoreti-cal basis for biological control of E. turcicum.

Please see our revised manuscript for details (Page 2, line 102-113)

Comments 2: The Conclusion section needs to be modified to point more specific to the most important findings of this study. In the current form is too general.

Response 2: Thanks for your kind comments. We have rewritten the conclusion.

The phyllosphere microbiome plays an indispensable role in plant health. Metogenomic sequencing was performed on the adaxial and abaxial surfaces of leaves from healthy and E. turcicum-inoculated maize. The results showed that E. turcicum altered the α and β diversity of the phyllosphere microbiome, and after maize was infected with E. turcicum, the composition and structure of the phyllosphere microbiome changed, with significant reductions in Shannon diversity and Chao1 index, as well as changes in the abundance of microbiome on both the adaxial and abaxial surfaces of leaves. It was found that microbial abundance and diversity were more correlated with the healthy maize phyllosphere, and more beneficial microorganisms and fewer pathogen accumulated on the CKWT phyllosphere, E. turcicum may have altered the balance between commensal and pathogenic microorganisms, which could have a positive impact on the microbiome's ability to regulate microbial functioning and maintain plant health. E. turcicum also altered the microbial abundance of disease resistance-related metabolic pathways and functional microbiome, and altered microbiome differences between the adaxial and abaxial surfaces of leaves. These findings are critical for further studies of the healthy maize phyllosphere microbiome for the isolation and screening of beneficial microorganisms that can enhance plant disease resistance, as well as for understanding the mechanisms by which E. turcicum affects the maize phyllosphere microbiome in terms of disease resistance and providing a theoretical basis for the biological control of E. turcicum.

Please see our revised manuscript for details (Page 18, line 478-496)

Reviewer 4 Report

Comments and Suggestions for Authors

Dear Authors,

I consider that the manuscript attempts to present the characterization of the maize leaf phyllosphere microbiome upon infection with E. turcicum, with very interesting results. However, some observations are made that could contribute to improve the manuscript.

Lines 7, 12, 16, 19, 20, and 22. Some dots at the end of the affiliations should be eliminated.

Introduction 

Line 46. Check the scientific name of maize according to https://wfoplantlist.org/

Line 48. It is not necessary to put the name of the pathogen in parentheses (E. turcicum).

Lien 50. There is an extra space between of and Northern. Please revised same situation in lines 94, 115, 143, 158. 

Lines 98-86. I consider that if examples are to be given, they should be as close as possible to the case under study.

Materials and Methods

Line 95. It should be School of Life Sciences.

Line 95. Was the collected soil treated? For example, the soil was sterilized? 

Line 96. It should be E. turcicum … Revise the same problem throughout the manuscript. 

Line 98. I consider that the process of inoculation of corn leaves should be explained in detail or refer to similar work. Can you explain how do you ensure that both leaf surfaces are inoculated with approximately the same amount of inoculum? What amount of inoculum is used?

Line 99. How is the sampling of both leaf surfaces for analysis performed? 

Lines 100-101. The meaning of CKWT and EWT should be explained as this is the first time it is mentioned in the manuscript beyond the abstract.

Lines 101-102. Was water always added to the soil?

The number of plants used and whether repetitions are carried out should be mentioned in the experiment. In addition to the code number, the maize cultivar used should be mentioned.

Line 106. Could you mention how it was determined that 10 g of microorganisms were collected?

Which of the leaves of the maize plants were used in the experiment? all or specific ones? When a maize leaf was used, only one or both surfaces of the leaf were analyzed. For example, in the case of CKWT-adaxial and CKWT-abaxial, only one or both surfaces were analyzed? 

Line 108. It should be 5 min, 1 min and 10 s. 

Line 121. Eliminate dot after (NCBI)

Line 129. It should be [24]. Community … Also, software. Relevant … 

Results

Line 140. Consider 3.1. E. turcicum … 

Line 151. Consider to include Figure 1 after this paragraph. 

Lines 178-170. I consider that the phyla mentioned should not be in italics. 

Line 170. It should be (Figure S2). Instead of (Figure S2), 

Line 205. Considerations are made that should be in the discussion only. Revise in the rest of Results. 

Line 208. Consider to include Figure 2 after this paragraph.

Line 214. Consider … album, and Salinomyces …

Figure 2 should be revised since Protoebacteria is considered as a bacterial genus. On the other hand, the ones that are considered as genus should be in italics.

Lines 231-232. Consider surfaces in both CKWT and EWT maize leaves

Line 234. I could not find anything about OE maize plants in Materials and methods.

Line 242. It should be reviewed because Basidiomycota is mentioned as a fungal genus, when it is not. This aspect should be revise in Figure 3. 

Line 301. sp. should not be in italics. Is recommended to check this aspect in the whole manuscript and figures and tables. 

Line 302. It should be M. aquaticum … Once the full name of the species is mentioned, it is not necessary to mention it again in the body of the manuscript. Check for other cases in the text.

Discussion 

In general, I recommended not to repeat some of the results already mentioned and not to refer to the tables and figures where they appear. Also, some emphasis should be made explicitly on how the results achieved contribute to the management of E. turcicum as stated in the title.

Author Response

Comments 1: Lines 7, 12, 16, 19, 20, and 22. Some dots at the end of the affiliations should be eliminated.

Response 1: Thank you for pointing this out. We have deleted the dot at the end of the affiliations.

Please see our revised manuscript for details (Page 1, line 7, 12, 16, 19, 20 and 22)

Comments 2. Introduction

Line 46. Check the scientific name of maize according to https://wfoplantlist.org/

Response 2: Thank you for pointing this out. We have revised the scientific name of maize.

Please see our revised manuscript for details (Page 2, line 49)

Comments 3. Line 48. It is not necessary to put the name of the pathogen in parentheses (E. turcicum).

Response 3: Thank you for pointing this out. We have deleted the name of the pathogen in parentheses.

Please see our revised manuscript for details (Page 2, line 52)

Comments 4. Line 50. There is an extra space between of and Northern. Please revised same situation in lines 94, 115, 143, 158.

Response 4: Thank you for pointing this out. We have deleted extra spaces.

Please see our revised manuscript for details (Page 2, line 53; Page 3, line 116; Page 4, line 152; Page 4, line 180, line 195).

Comments 5. Lines 98-86. I consider that if examples are to be given, they should be as close as possible to the case under study.

Response 5: Thanks for your careful comments.

This paragraph mainly describes the effects of pathogens on the plant phyllosphere microbiome. We consulted relevant study and added it, but we did not find any studies on the specific analysis of the effects of pathogens on the microbiome of the adaxial and abaxial leaves of maize, and could only describe relevant studies in other species.

Foliar infection of maize plants by S. turcica can induce the enrichment of beneficial bacterial communities in the maize rhizosphere, enhancing the plant's defense against pathogen infection (Zhu et al, 2021). The Shannon and Simpson indices of the phyllosphere bacterial communities in highly resistant hybrid maize are significantly higher than those in susceptible one (Tian et al, 2020).

Please see our revised manuscript for details (Page 2, and line 87-91).

References:

Zhu L, Wang S, Duan H, Lu X. Foliar pathogen-induced assemblage of beneficial rhizosphere consortia increases plant defense against Setosphaeria turcica. Front Biosci (Landmark Ed). 2021 Sep 30;26(9):543-555. doi: 10.52586/4966.

Xueliang T, Dan X, Tingting S, Songyu Z, Ying L, Diandong W. Plant resistance and leaf chemical characteristic jointly shape phyllosphere bacterial community. World J Microbiol Biotechnol. 2020 Aug 16;36(9):139. doi: 10.1007/s11274-020-02908-0.

Comments 6: Materials and Methods

 Line 95. It should be School of Life Sciences.

Response 6: Thank you for pointing this out. We have revised.

Please see our revised manuscript for details (Page 3, line 117)

Comments 7: Line 95. Was the collected soil treated? For example, the soil was sterilized?

Response 7: Thank you for pointing this out. The soil wasn’t treated. It was directly excavated from the field, mixed and placed in each pot to ensure consistency in the growth environment.

Please see our revised manuscript for details (Page 3, line 119)

Comments 8: Line 96. It should be E. turcicum … Revise the same problem throughout the manuscript.

Response 8: Thank you for pointing this out. We have revised.

Please see our revised manuscript for details (Page 3, line 120; Page 17, line 396, line 400)

Comments 9: Line 98. I consider that the process of inoculation of corn leaves should be explained in detail or refer to similar work. Can you explain how do you ensure that both leaf surfaces are inoculated with approximately the same amount of inoculum? What amount of inoculum is used?

Response 9: Thanks for your careful comments.

Lab-preserved E. turcicum strains were used, which were transferred from glycerol stored at −80 °C to PDA medium and cultured at room temperature for 2–3 weeks. The spore suspension of E. turcicum was prepared by flooding each PDA plate with ap-proximately 8 mL of ddH2O, which contained 0.1% Tween 20. The spores were then gently removed using a glass rod. Subsequently, approximately 5 mL of the undi-luted spore suspension was transferred into each centrifuge tube. To adjust the concen-tration of the suspension, additional ddH2O was added until it reached a final concen-tration of 4 × 103 conidia per ml (Zhang et al, 2020). An artificial inoculation procedure was carried out as described by Wu et al. (Wu et al, 2014). Spray 1ml of spore suspension of E.turcicum on both the adaxial and abaxial surfaces of the leaves at the seedling stage (V4), respectively.

Please see our revised manuscript for details (Page 3, and line 120-129).

References:

Zhang, X.; Fernandes, S.B.; Kaiser, C.; Adhikari, P.; Brown, P.J.; Mideros, S.X.; Jamann, T.M. Conserved defense responses between maize and sorghum to Exserohilum turcicum. BMC Plant Biol. 2020, 20, 67. https://doi.org/10.1186/s12870-020-2275-z.

Wu, F.; Shu, J.; Jin, W. Identification and validation of miRNAs associated with the resistance of maize (Zea mays L.) to Exserohilum turcicum. PLoS ONE. 2014, 9, e87251. https://doi.org/10.1371/journal.pone.0087251.

Comments 10: Line 99. How is the sampling of both leaf surfaces for analysis performed?

Response 10: Thank you for pointing this out.

The sampling time and position of several samples were the same. The leaves were the fifth and sixth leaves counted from bottom to top. The two surfaces of the leaves were from the same position on the same leaf of the same seedling. As you suggested, the procedure of collect samples from both adaxial and abaxial surfaces was clearly rewritten for a better understanding of the details in this experiment.

Please see our revised manuscript for details (Page 3, and line 132-135).

Comments 11: Lines 100-101. The meaning of CKWT and EWT should be explained as this is the first time it is mentioned in the manuscript beyond the abstract.

Response 11: Thank you for pointing this out.

CKWT represented wild maize without inoculation with E. turcicum, EWT represented wild maize inoculated with E. turcicum.

Please see our revised manuscript for details (Page 3, and line 136-137).

Comments 12: Lines 101-102. Was water always added to the soil?

Response 12: Thank you for pointing this out.

Seedlings were watered once or twice per week, depending on soil humidity, 500ml each time.

Please see our revised manuscript for details (Page 3, and line 139).

Comments 13: The number of plants used and whether repetitions are carried out should be mentioned in the experiment. In addition to the code number, the maize cultivar used should be mentioned.

Response 13: Thank you for pointing this out.

Three replicates for each group and three seedlings for each replicate.

Please see our revised manuscript for details (Page 3, and line 138).

 The maize cultivar is KN5585, it was in Page3, line 116.

Comments 14: Line 106. Could you mention how it was determined that 10 g of microorganisms were collected?

Response 14: Thank you for pointing this out.

Our description was not quite correct, the sentence has been rewritten as “A sterile cotton swab was used to randomly scrape microorganisms from the surface of 10g fresh leaves into a centrifuge tube and added 90 ml of sterile 0.1 mmol/L potassium phosphate buffer solution (pH=7.4).

Please see our revised manuscript for details (Page 3, and line 143).

Comments 15: Which of the leaves of the maize plants were used in the experiment? all or specific ones? When a maize leaf was used, only one or both surfaces of the leaf were analyzed. For example, in the case of CKWT-adaxial and CKWT-abaxial, only one or both surfaces were analyzed?

Response 15: Thank you for pointing this out.

The leaves were the fifth and sixth leaves counted from bottom to top. The two surfaces of the leaves were from the same position on the same leaf of the same seedling. When a maize leaf was used, both surfaces of the leaf were analyzed. CKWT-adaxial and CKWT-abaxial come from two surfaces of the same leaf.

Please see our revised manuscript for details (Page 3, and line 132-135).

Comments 16: Line 108. It should be 5 min, 1 min and 10 s.

Response 16: Thank you for pointing this out. We have revised.

Please see our revised manuscript for details (Page 3, line 146)

Comments 17: Line 121. Eliminate dot after (NCBI)

Response 17: Thank you for pointing this out. We have revised.

Please see our revised manuscript for details (Page 4, line 158)

Comments 18: Line 129. It should be [24]. Community … Also, software. Relevant …

Response 18: Thank you for pointing this out. We have revised.

Please see our revised manuscript for details (Page 4, line 166, line 170)

Comments 19: Results.

Line 140. Consider 3.1. E. turcicum …

Response 19: Thank you for pointing this out. We have revised.

Please see our revised manuscript for details (Page 4, line 176)

Comments 20: Line 151. Consider to include Figure 1 after this paragraph.

Response 20: Thanks for your kind comments. We have added.

Please see our revised manuscript for details (Page 4, line 188)

Comments 21: Lines 178-170. I consider that the phyla mentioned should not be in italics.

Response 21: Thanks for your kind comments. We have revised.

Please see our revised manuscript for details (Page 5, line 205-207)

Comments 22: Line 170. It should be (Figure S2). Instead of (Figure S2),

Response 22: Thanks for your kind comments. We have deleted.

Please see our revised manuscript for details (Page 5, line 207)

Comments 23: Line 205. Considerations are made that should be in the discussion only. Revise in the rest of Results.

Response 23: Thank you for pointing this out. We have revised.

Please see our revised manuscript for details (Page 7, line 242; Page 13, line 322)

Comments 24: Line 208. Consider to include Figure 2 after this paragraph.

Response 24: Thanks for your kind comments. We have added.

Please see our revised manuscript for details (Page 7, line 245)

Comments 25: Line 214. Consider … album, and Salinomyces …

Response 25: Thank you for pointing this out. We have revised.

Please see our revised manuscript for details (Page 7, line 251)

Comments 26: Figure 2 should be revised since Protoebacteria is considered as a bacterial genus. On the other hand, the ones that are considered as genus should be in italics.

Response 26: Thank you for pointing this out. There was an error in our labeling, as it does not have a specific classification at the genus level, but belongs to the Protoebacteria at the phylum level. We have revised the Figure 2.

Please see our revised manuscript for details (Page 10, line 255)

Comments 27: Lines 231-232. Consider surfaces in both CKWT and EWT maize leaves

Response 27: Thank you for pointing this out. We have revised.

Please see our revised manuscript for details (Page 8, line 269-270)

Comments 28: Line 234. I could not find anything about OE maize plants in Materials and methods.

Response 28: Thank you for pointing this out. This is a mistake in our writing, it should be CKWT and EWT here, and the sentence was “To identify the differences in bacterial and fungal communities between the two leaf surfaces of OE CKWT and EWT maize, 12 dominant bacterial and fungal groups (with relative abundance≥1.0%) at the species level were detected (Figure 3)”.

Please see our revised manuscript for details (Page 8, line 271-273)

Comments 29: Line 242. It should be reviewed because Basidiomycota is mentioned as a fungal genus, when it is not. This aspect should be revise in Figure 3.

Response 29: Thank you for pointing this out. There was an error in our labeling, as it does not have a specific classification at the genus level, but belongs to the Basidiomycota at the phylum level. We have revised the Figure 3.

Please see our revised manuscript for details (Page 12, line 290)

Comments 30: Line 301. sp. should not be in italics. Is recommended to check this aspect in the whole manuscript and figures and tables.

Response 30: Thank you for your detailed suggestions, which have greatly helped improve the rigor of the manuscript. In response to this formatting problem, we carefully reviewed the whole manuscript and made revisions one by one.

Please see our revised manuscript for details (Page 13, line 343, line 349; Page 14, line 356, line 358, line 359, line 360)

Comments 31: Line 302. It should be M. aquaticum … Once the full name of the species is mentioned, it is not necessary to mention it again in the body of the manuscript. Check for other cases in the text.

Response 31: Thank you for pointing this out. We have revised.

Please see our revised manuscript for details (Page 13, line 346; Page 14, line 358)

Comments 32: Discussion

In general, I recommended not to repeat some of the results already mentioned and not to refer to the tables and figures where they appear. Also, some emphasis should be made explicitly on how the results achieved contribute to the management of E. turcicum as stated in the title.

Response 32: Thanks for your careful comments. We have deleted repeated results and added how the results contribute to the management of E. turcicum. We also rewtitten the conclusion to highlight the contribution of our findings and findings to the biocontrol of the E. turcicum.

Please see our revised manuscript for details (Page17, and line 404-408; Page18, line 430-432, line 434-439, line 452-453, line 458-461, line 478-496).

Round 2

Reviewer 4 Report

Comments and Suggestions for Authors

Dear Authors,
I consider that the manuscript presents very interesting results on cow changes in the microorganisms of the phyllosphere may be related to the defense response of maize plants. Responses to the observations made were well conducted, although there are some details that could be improved.

Line 49. L. should be not in italic. 
Lines 127-128. It should be mL instead of ml. Always separate the numbers from the units of measurement except for %. Revise the entire document. 
Line 144. Use always the international system of measurements. It should be mmol L-1 instead of mmol/L.    
Line 349. It should be Rugosimonospora africana   

Author Response

Comments 1: Line 49. L. should be not in italic.

Response 1: Thanks for your careful comments. We have revised.

Please see our revised manuscript for details (Page 2, line 49)

Comments 2: Lines 127-128. It should be mL instead of ml. Always separate the numbers from the units of measurement except for %. Revise the entire document.

Response 2: Thanks for your careful comments. We have revised.

Please see our revised manuscript for details (Page 3, line 127, line 139, line144, line148)

Comments 3: Line 144. Use always the international system of measurements. It should be mmol L-1 instead of mmol/L.

Response 3: Thanks for your careful comments. We have revised.

Please see our revised manuscript for details (Page 3, line 144)

Comments 4: Line 349. It should be Rugosimonospora africana.

Response 4: Thank you for pointing this out. We have revised.

Please see our revised manuscript for details (Page 13, line 348)